# Can the Theory of Salutogenesis Offer a Framework to Enhance Policy Coherence during Policy Development and Implementation in Municipalities?

**Ruca Maass * and Monica Lillefjell**

Faculty of Medicine and Health Sciences, Norwegian University of Science and Technology, Tungasletta 2, 7047 Trondheim, Norway; monica.lillefjell@ntnu.no
* Correspondence: ruca.e.maass@ntnu.no

**Abstract:** Health promotion efforts in municipalities demands extensive collaboration, between sectors (multi-sector) as well as between levels (multi-level). To develop, adopt and implement policies that contribute to reduce health inequity demands for policy coherence: consistent approaches across sectors and levels. In this conceptual paper, we discuss how the theory of Salutogenesis, and its inherent concept of coherence, can contribute to structure such collaboration processes in coherent ways, and contribute to experiences of policy coherence for both collaborators (practitioners and stakeholders) and inhabitants. We discuss how challenges with fragmented knowledge as well as lack of resources and motivation might be met through an explicit application of Salutogenesis core concepts. First, we argue that framing HP-efforts in terms of positive goals that can be achieved can contribute to motivation for change. Next, we discuss how the concepts of comprehensibility, manageability and meaningfulness can be applied to describe challenges, and plan for coherent policies and processes vertically (between levels) and horizontally (between sectors). Last, we discuss limitations and challenges with this approach, including the need to further develop the theoretical foundation of Salutogenesis, and how it can be applied at a setting- and/or policy-level.

**Keywords:** Salutogenesis; multi-sector; collaboration; health promotion; municipalities; policy coherence

## 1. Introduction

Health promotion (HP) aims at improving the upstream-conditions for health and health equity through addressing the settings of everyday life where people "live, work, love and play" [1,2]. This emphasizes the key role local governments and municipalities play in respect to population health. Applying a settings-approach unto the municipal context implies to focus on the physical, organisational, and social contexts in which people live their lives [3,4]. This includes to address health and conditions for health that lie outside the traditional "health"-sector, as expressed in a "health all policies" (HiAP)-approach [5]. Accordingly, in order to promote health, health-relevant issues must be addressed across municipal sectors and settings (such as schools, work-places, NGO's and social groups). Municipalities might thereby be described as super-settings, in which distinct settings interact and create experiences in, across and in-between these distinct settings [6].

To achieve HiAP in a local context, efforts have to be planned and implemented in a co-ordinated manner across sectors and actors, to ensure that they contribute to fulfill the overarching aims, such as promoting health and well-being, or to reduce social inequality in health [6]. Challenges can arise when policies contradict each other across or within settings; or if measures are implemented and carried out in ways that contradict the aim they are meant to fulfill [7,8]. Accordingly, to achieve HiAP and promote population health locally implies extensive collaboration and cooperation processes within and between local (municipal) actors and sectors. This is also linked to the growing realization that sector-specific policies often are too narrow to address underlying causes of specific challenges [9];

while framing societal challenges as "wicked problems" might either contribute to paralysis (as the challenges seem overwhelming), or to an over-estimation about what can be done about these challenges [9].

All in all, coordinating policies and efforts horizontally and vertically emerges as a crucial strategy to overcome challenges and develop relevant solutions for better population health [9,10]. Termeer and colleagues [9] stress the importance of interaction with stakeholders to ensure legitimacy of policies, especially in the early stages of the policy cycle, when the agenda is set and goals are defined. However, how policies are set out into life, and how they are received and work in the local context also depends on processes linked to formulation, adoption and implementation. Especially involving target groups is crucial to ensure relevant, legitimate policies that contribute to health equity [7,10]. To ensure support for policies and solutions from both the population, practitioners and decision-makers, stakeholder- and public involvement throughout the whole process of planning, implementing and evaluating policies seems beneficial [11]. However, multi-sectoral and -level collaboration linked to the development and implementation of policies is often challenged by issues linked to motivation, fragmented knowledge, understandings and responsibilities, as well as rigorous and sector-specific input/output regimes and a lack of resources to follow up on responsibilities across sectors and levels [12–17].

Even if policy coherence thereby emerges as a central goal and mean for health promotion in municipalities, no sound theoretical framework to describe, assess or promote coherence at a setting- or policy-level has been developed yet. In this conceptual paper, we want to explore how the theory of Salutogenesis and the concept of sense of coherence can be applied to increase policy coherence during the development, planning and implementation of HP policies in municipalities.

### 1.1. Policy Coherence

Policy coherence describes the degree to which policies establish consistent aims and strategies to pursue this aims across government department, agencies and actors and is explicitly linked to health equity [17,18]. A lack of policy coherence can result in ineffective or even counter-acting efforts across sectors and levels. Worse, policy incoherence might even damage for perceptions of legitimacy, and undermine motivation to engage in collaboration and implementation processes among stakeholders and the population [7,17].

Fragmented knowledge, output-driven budgets and sector-specific goals and responsibilities can lead to fragmentation, and result in "silo"- or "stovepipe" thinking that spoils for policy coherence and collaboration-processes during the whole policy-cycle of developing, adopting, planning and implementing policies. Examples of policy incoherence span from attempts to pursuit valid goals with irrelevant or even counter-induced means; expectations that are raised but not met; or if the goals and actions from one sector are contradictive for the goals and activities of other sectors [7,17,18]. Challenges during collaboration processes, including fragmented knowledge and responsibilities, can add to policy incoherence, and result in incoherent experiences for both collaborators and the population.

Policy incoherence can have major impacts on population health and health equity, as incoherencies often become visible in the experiences of minorities and marginalized groups [18]. A recent example of a minority group experiencing policy incoherence has become visible in the "Black lives matter!"- movement. The police are a crucial societal resource that should protect citizens and their rights from violations. However, these expectations are not in line with the experiences that people of colour make in contact with the police in the USA: instead of being a resource and protector, the police are often perceived as a threat and/or a violator themselves. This might be due to conflicting policies (such as establishing quantitative goals for arrestations simultaneously as setting goals for more communication), or at incoherencies during implementation (for example, police officers not receiving adequate education, lack time to negotiate or experience pressure to be "tough" from supervisors that can come in conflict with their responsibility to "protect everybody's lawful rights"). If expectations towards societal resources are not supported

by personal experiences; if valid goals are pursuit with means that do more damage than good; or if marginalized groups are instrumentalized rather than heard in public policy development despite aspirations of including processes; incoherencies in the experience of citizens, practitioners and stakeholders occur.

Simultaneously, when it comes to health promotion and health equity, "what" we do might be less important than "how" we do it: even if everyone agrees on shared goals, might differing understandings and positions, applicable resources to aspire these goals and personal or professional values and interests hazard for coordinated efforts, and contribute to perceptions of incoherence during collaboration for stakeholders, as well as incoherent outcomes for inhabitants. Thus, matters of coherence are crucial during developing, planning, and implementing policies, and can have major impact on whether a policy works in line with intentions.

Accordingly, the WHO has established "policy coherence" as a valid goal for health promotion and a mean to achieve health equity: "Coherence across all areas of public policy is important to realize health equity and well-being for all" [18]. Incoherence might occur horizontally and vertically: Challenges with horizontal coherence refer to the above-described silo-thinking, indicating different understandings, values, and strategies. Challenges with vertical coherence can describe processes within sectors, from decision-makers over implementers and practitioners to inhabitants and vice versa.

*1.2. Salutogenesis and the Sense of Coherence*

Salutogenesis describes health independent from illness, along a gradient ranging from ill health to perfect health [19,20]. According to its founder Aaron Antonovsky, it is the individual "sense of coherence" (SOC) that determines our location on the health-gradient. A strong SOC pushes us towards the healthy end of the continuum. A strong SOC implies to perceive the world as comprehensive, manageable, and meaningful. SOC is developed through the internalization of resources and experiences that touch into these three dimensions of coherence simultaneously [20]. Thus, Salutogenesis offers a theoretical framework that (a) links perceptions of coherence to health outcomes [21,22] and (b) describes how coherence is experienced at the individual level [19,20]. Even if we do not know (yet) if and how this conceptualization can be translated unto settings or processes, does Antonovsky point out that the individual sense of coherence is developed through experiences with societal structures and resources. In turn, one might ask what kind of societal structures and processes do promote coherent experiences- and how these are linked to coherent policies.

According to Salutogenesis are perceptions of comprehensibility, manageability, and meaningfulness linked to distinct experiences, that provide pinpoints towards aspects that can enhance perceptions of coherence- and thereby, might serve as starting points to explore "coherence" at higher levels. For example, *comprehensibility* is linked to consistent communication from above in matters that cannot be influenced easily, to enable individuals to "discover structure" and learn the rules of the game [20,23]. On the other hand, possibilities to engage in power-equal inter-group dialogue also adds to comprehensibility by adding individual or group-wise experiences into the common narrative; and by offering opportunities to contextualize experiences with society and societal resources beyond personal experiences [15,23]. This allows for learning more about how they work for "others" and prepares for more flexible use later. *Manageability* is linked to experiences of load-balance, that is; having enough over-come able challenges and adequate resources to resolve these. *Meaningfulness* is closely linked to participating in shaping outcomes and refers to perceptions of being an active agent in one's own life [20].

While strategic approaches like the super-settings approach and HiAP demand for establishing common goals and engage in co-ordinated collaboration across sectors and actors, they do not provide a sound framework about how such processes can be structured in a coherent manner. In the below discussion, we want to explore if the concept of coherence, as described in Salutogenesis, offers a framework to assess and promote

coherence during the development, adoption, planning and implementation of HP policy in municipalities.

## 2. How Can Salutogenesis Contribute to Structure Collaboration Processes?

Salutogenesis aims to provide HP with an overarching theoretical framework that focuses on resources, assets and positive outcomes; with a strong emphasize growing with challenge and learning processes [24]. It also establishes an operationalization of the term "sense of coherence" along the three dimensions of comprehensibility, manageability, and meaningfulness, and link these to distinct experiences.

However, to be a beneficial theory for HP, Salutogenesis needs to expand it's scope, and develop knowledge on how salutogenic, coherent experiences can be facilitated through societal settings, processes and policies. During this discussion, we want to explore how Salutogenesis can be applied to (a) set the agenda and define desirable outcomes for policies and (b) contribute to experiences of vertical and horisontal coherence during the collaboration processes during the adoption and implementation of HP policy.

### 2.1. Salutogenic Outcomes

Applying Salutogenesis in its widest sense means to look for health-promoting factors and processes, and support these [24,25]. The focus is on enhancing the positive, developing resources and achieving positive outcomes. This clearly distinguishes health promotion from preventive efforts, which aim at reducing negative influences and consequences- and is thus linked to a pathogenic line-of-thought. Applying an explicit salutogenic focus means to turn the focus from "wicked problems" unto "co-benefits", and phrase positive goals that can be achieved through health promotion efforts. Working towards positive goals can contribute to motivation directly and might be a beneficial starting point to define roles and responsibilities as this is not linked to implying earlier "failure" or "guilt". Phrasing positive goals linked to increased health, life satisfaction or social relationships might also help to overcome the role as "fire-extinguisher", rushing from challenge to challenge, and enable practitioners to act as visionaries for a positive society-development. The SDG are a major step towards making inter-connected benefits visible through defining overarching goals for a positive society-development [26]. However, even if these represent common aspirations, it remains unclear how these goals can be pursued from specific sectors and actors. Translating overarching goals into practical, sector- specific, positively framed aims might help to clarify roles and responsibilities—which is crucial for transforming intentions into actions.

Moreover, as SOC emerges as a crucial health resource, "strengthen SOC in the population" emerges as a valid goal for HP-measures in its own right [27]. This implies to offer opportunities to achieve a more comprehensive, manageable, and meaningful outlook at the world from various starting points. Different positions on the three dimensions facilitate for different experiences [20] and might call for different strategies and approaches. For example, supporting individuals or groups that are high on comprehensibility (having an accurate picture of one's situation) and meaningfulness (high motivation to improve one's situation), but low in manageability (having the resources to move from where you are to where you want to go) suggests approaches that aim to redistribute resources and/or challenges, and are linked to social equity. However, a starting point characterized by low comprehensibility, but high manageability and meaningfulness might demand for more accurate knowledge to enable individuals and communities to apply their resource in adequate ways- which bring them closer to their desired outcomes. Last, meaningfulness emerges as "the most important dimension" of SOC [20]. As meaningfulness is closely linked to experiences of participation in shaping outcomes, establishing participatory processes and inclusive decision-making as an important mean and desired outcome for local HP. This is closely linked to vertical coherence, which is about anchoring aims and approaches at all levels of the involved sectors- from decision-makers over practitioners to inhabitants and vice versa.

*2.2. Vertical Coherence: Involvement in Bottom-Up/Top-Down Processes*

One important aspect of any HP-measure is that it must be well-received and adequately applied by the public to have any effect on population health. However, well-intended measures can be spoiled if solutions are perceived as little relevant; if design and/or contextual matters spoil for the benefits of the proposed solution; or if experiences during the development and implementation of the measure contribute to perceptions of incoherence for inhabitants. Especially, expectations about participation in shaping outcomes that are not met during the implementation process can spoil for the expected benefits of inclusive approaches, such as empowerment, trust and an increased sense of coherence. The same might be true for collaborators at lower levels, which might be expected to carry out the specific actions and activities within their sector, but do not receive additional resources to resolve the challenges they are presented with "top-down".

Thus, challenges for vertical coherence can be summarized in line with the three dimensions of coherence (see Figure 1). Challenges with comprehensibility can arise in respect to insecurity about how to address the challenge, but also in respect to the decisions- and implementation process itself. This includes knowledge about how and when to participate to what means, how contributions are followed-up and knowledge about contextual matters that set the boundaries for actions (such as available resources in the municipality). Challenges with manageability arise in respect to finding the resources to participate in and/or carry out activities; and in respect to how opportunities for involvement are designed and communicated especially to people and groups in vulnerable situations. Challenges with meaningfulness manifest themselves as lack of motivation to participate, or to use the developed resource after the implementation is concluded. Motivation is often spoiled by perceptions of not being able to make a real difference, either because contributions are not followed up or because one does not see the relevance of the measure or activity; or because one lacks information to make sound contributions.

To plan for vertical coherence during implementation processes implies to actively apply bottom-up and top-down approaches that are integrated in a coherent manner.

For example, comprehensiveness is linked to information about the coming process(es), including opportunities to participate and influence outcomes. This information should be provided by local authorities in a consistent manner to enable inhabitants to discover the structure of the process and contribute in coherent ways. Simultaneously, facilitating for knowledge-sharing bottom-up processes can broaden understandings, and ensure that measures are implemented in a way that is supported by the local community. Manageability during implementation can be addressed through offering low-threshold opportunities for participation and facilitate for joined learning activities to ensure that all stakeholders, including inhabitants, have relevant knowledge to contribute to the development of realistic measures. Moreover, involving inhabitants and local practitioners in defining goals and desirable outcomes might add to manageability in the long run, if applied measures will result in relevant resources that can be applied to resolve later challenges. Being involved in defining desirable outcomes can also enhance motivation during collaboration, as well as perceived relevance of the developed measure/resource. Thus, while central goals and aspiration can give pinpoints about desirable outcomes; establishing local aims anchored in the local community is crucial to enhance meaningfulness and ensure motivation to participate. Moreover, applying and communicating predictable, committing processes during implementation can make contributions visible, and might contribute to build trust and facilitate for empowerment through increasing security in respect to how contributions are handled.

**VERTICAL INCOHERENCE**

*Challenging Comprehensibility*

- What is going on- what is planned?
- How and when can I contribute?
- Do I know enough to make sound contributions?

*Challenging Manageability*

- Do I have the time and resources to participate?
- Are opportunities for participation designed in an inclusive way?
- Will participation increase manageability in the long run by providing new, adequate resources?

*Challenging Meaningfulness*

- Does the measure seem relevant for my needs and aspirations?
- Was I involved in mapping out our goals, and my role in achieving them?
- How will my contributions be handled- will I make a difference?

**Figure 1.** Vertical incoherence.

*2.3. Horizontal Coherence: Multi-Sector Collaboration Processes*

Challenges in multi-sectoral collaboration are often summarized as matters of "silo-thinking", characterized by fragmented knowledge, goals, and responsibilities that damage for motivation and poses obstacles for collaboration processes [15]. In a way, a lot of these challenges could be described as "incoherence" (see Figure 2); according to the salutogenic framework.

HORZONTAL INCOHERENCE

***Challenging Comprehensibility***

- Is this my problem?
- How do I understand the problem? How do others?
- How can I contribute?
- What do I need to know to contribute in meaningful ways?
- What do others need to know to understand my role, needs and valid goals?

***Challenging Manageability***

- Resources are linked to measurable outcomes (NPM)
- Do I have the resources to contribute?
- Can I access these resources?
- Will my contribution be visible?
- Will my contribution lead to more or fewer resources in the long run?

***Challenging Meaningfulness***

- Do I see the value of what we are doing?
- Do I believe what we try to achieve is important?
- Can I contribute in meaningful ways?
- Will my contribution contribute to achieve my goals- or am I investing in other's goals?
- Will I receive feedback on my contribution?
- Was I involved in mapping out our goals, and my role in achieving them?

**Figure 2.** Horizontal incoherence.

For example, fragmented knowledge and understandings about desired outcomes might be described in terms of comprehensibility. Differing understandings about health and the determinants of health can yield obstacles for establishing a common ground as well as for outlining the specific roles and responsibilities assigned to involved stakeholders. Confusion about what you are supposed to do, and how your efforts can contribute to the common goal might likewise represent challenges with comprehensibility. A focus on comprehensibility during implementation implies to give all stakeholders sound understandings about what we want to achieve (good population health, social equity) and how we want to achieve it (addressing the social determinants). Receiving input and knowledge about central aspects and mechanisms might help to outline one's own professional role in the collaboration. Moreover, including joined learning activities and knowledge-sharing enables processes that draw on the specialized knowledge of collaborators, and contributes to better-informed decisions and motivation among stakeholders [16,17,28] Information-sharing both between sectors and levels is a crucial step to establish a common ground and ensure that stakeholders find meaningful perspectives that enables them to work towards the common goal from their specific position and represents a chance to increase understanding about different roles and responsibilities across stakeholders [2].

Manageability in collaboration processes is often linked to the distribution of resources between stakeholders; both between sectors and levels. Being assigned new responsibilities and tasks, but not receiving the funds, time or competencies needed to fulfil this role spoils for manageability and poses major obstacles for effective implementation of measures. Thus, when specific sectors, levels or actors are assigned responsibilities, these should be accompanied by relevant resources to ensure manageability. As post-new public management strategies include to distribute fundings according to earlier achievements, it should be ensured that contributions to common goals are made visible at the point (sector and level) from which they are addressed [28]. Thus, measurable outcomes at each level and in each sector would be defined to make contributions visible.

Last, perceptions of meaningfulness are crucial for motivating stakeholders to participate in collaboration processes. Matters of meaningfulness are linked to perceptions of contributing to reach important goals, seeing the value of one's own contribution, being able to make contributions visible for others, and receiving adequate feedback. Most important, according to theory is meaningfulness linked to participation in shaping outcomes [20,23]. This emphasizes the importance of inclusive processes throughout the collaboration; from defining goals to implementing approaches. Moreover, anchoring and translating overarching goals in the specific plans and strategic documents for each involved sector can contribute to make sector-specific contributions visible, which can increase motivation and perceptions of working to achieve one's own goals simultaneously as one contributes to a greater purpose- which is at the heart of collaboration. Last, phrasing positive goals that can be achieved can contribute to highlight co-benefits of approaches, and turn the focus from the never-ending wicked problems to engaging in joined learning and positive development processes.

## 3. Reflections

Policy coherence emerges as a valid goal and central strategy for health promotion and tackling health inequity municipalities. Throughout this article, we argue that the theory of Salutogenesis might offer a beneficial framework to outline how to describe, assess, and plan for policy coherence and experiences of coherence during collaboration in a municipal context.

Applying Salutogenesis and its inherent operationalization of how "coherence" is experienced at a policy- and setting-level might yield obstacles: for once, sense of coherence is described as a relational concept, and with a strong focus on the individual and the active process of creating coherence at an individual level. Individual differences, values and prior held beliefs might make it difficult to translate the dimensions of how coherence is experienced unto how it can be described and promoted at higher levels. As coherence is something that emerges between the individual and the supporting settings, focusing on coherence solely at a setting-level might imply to assess coherence in line with majority-views, and thereby, even decreasing coherence for minorities and people in marginalized situations. Thus, a strong focus on involvement and participation especially from people in vulnerable situations is central, both in terms of ensuring the meaningfulness- dimension of coherence, and for enabling practitioners to build coherence bottom-up, based on the various experiences of different social groups and individuals.

Moreover, as indicated above, approaches that aim at increasing coherence and experiences of coherence need to consider the various starting points, and carefully choose strategies that address specific/local challenges with coherence. For example, a community having all the resources and being highly motivated for change but lacking a sound idea of what the challenge is and how it might be resolved need to apply other strategies for HP than a community that has a clear picture about what they want to achieve but lacks the resources to do so. In this context, Salutogenesis might not be sufficient to outline how to address specific local issues. Other approaches, such as for example health literacy, distribution of resources and social equity and/or empowerment might offer more in-depth considerations about how to work with specific challenges. However, the benefit

of applying Salutogenesis as an overarching framework might be to assess starting points, and design approaches that increase coherence through targeted interventions and coherent experiences while trying to pursuit these.

**Author Contributions:** Both authors have contributed equally to developing the framework presented and discussed in this article, and contributed to writing, editing and critically discussing the article as it took form. All authors have read and agreed to the published version of the manuscript.

**Funding:** This research received no external funding.

**Institutional Review Board Statement:** Not applicable.

**Informed Consent Statement:** Not applicable.

**Data Availability Statement:** Not applicable.

**Conflicts of Interest:** The authors declare no conflict of interest.

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
