# Peer review of "Can the Theory of Salutogenesis Offer a Framework to Enhance Policy Coherence during Policy Development and Implementation in Municipalities?"

_societies, doi:10.3390/soc12010024_

Round 1

Reviewer 1 Report

This article poses interesting ideas but it is built from a major  inconsistency that can be easily solved:

"sense of coherence" (the main theoretical approach of salutogenesis) should not be desiccated into "sense" + "coherence". This represents a misunderstanding of the concept and therefore when referring to "coherence" in the article it should be avoided this concatenation. The suggestion is to replace "coherence" with  "concinnity" (e.g.) when not referring to the specific concept of the sense of coherence: (ex) "...vertical coherence can be summarized in line with the three dimen- 204 sions of coherence..."   preferable "...vertical concinnity can be summarized in line with the three dimen- 204 sions of coherence..."

please review some typos in the text (ex are presented bellow)

its inherent concept 12 of coherence, can contribute to structure such collaboration processes in coherent ways, and con- 1

vertical coherence can be summarized in line with the three dimen- 204 sions of coherence. 

Policy coherence emerges as a valid goal and central strategy for health promotion 348 and tackling health inequity municipalities.

Applying Salutogenesis, and its inherent operationalization of coherence, at a policy- 353 and setting-level might yield obstacles:

for once, “coherence” in Salutogenesis is described 354 as a relational concept, and with a strong focus on the individual and the active process 355 of creating coherence at an individual level

Author Response

Dear reviewer 1,

Thank you for your kind and valuable feedback on this conceptual paper. You recommend to not apply the word "coherence" to describe the degree of internal consistency during collaboration processes, as not to confuse this concept with the "sense of coherence" (SOC) as described in Salutogenesis.

However, for us, it is the main point if this article to explore if and   how the concept of coherence, as it is described in Salutogenesis, can be applied at levels beyond the individual. The argument we make is that the thoroughly-described concept of "sense of coherence" builds on distinct (coherent) experiences, characterized by comprehensibility (linked to information and knowledge), manageability (linked to resources to resolve challenges) and meaningfulness (participation in shaping outcomes). The main question we discuss is whether this entry point yields benefits for the planning and conducting of collaboration process- that seek to provide collaborators with coherent experiences during, and the public with opportunities for coherent experiences after the collaboration. 

In other words, to explore the concept of coherence in line with how it is defined in Salutogenesis is the main aim of this paper, and we think it would be confusing to introduce another term. To meet your concerns, we have tried to clarify our focus on application of the theory in the introduction section, to clarify the distinctions made between sense of coherence (in the individual) and "coherence" as a characteristic of processes. We hope that this makes our intention clearer, and resolve some of the challenges you uncovered. 

Reviewer 2 Report

The article aims at considering if salutogenesis offer a framework to structure multi sectoral collaboration for health promotion in municipalities, which is a very interesting topic.

This article is a concept paper, without any data collection, case study or method.

While the aim of the paper is of interest, I have difficulties in seing how the authors articulate their article in regard to the litterature. In other words, there are mentions of settings based approach, of health in all policies, of health promotion, but not clear position about why salutogenesis is the most appropriate framework to contribute to structure collaboration process.

Moreover, on the other side, authors refers sometimes to policies, sometimes to collaboration process, without explicitly mentionning what is the focus in the article and do not position their paper in the policy cycle or in regard to policy sciences, as how salutogenesis could be crossed and helpfull in regard to this.

I would like to have seen more example of others studies questionning intersectorality, interdisciplinarity when its comes to local partnerships about health, and have more insights on how they are meaningfull for the salutogenesis.

Finally, the article lacks of structures and steps on how to use salutogenesis not just as asking questions or reflect on a process, but on how these questions should be used, at which stages, by whom, to provide full insights on its usefullness.

Author Response

Dear reviewer,

Thank you for your kind evaluation and valuable inputs regarding our article on whether the theory of Salutogenesis offers a framework to structure multi-sectoral collaboration for health promotion in municipalities. We have tried to improve the article in line with your suggestions. Please see our answer below for a more detailed description on how we handled your specific inputs.

First, as you point out, "This article is a concept paper, without any data collection, case study or method", but with the intention to explore a theoretical framework and how it might be applied and further developed in the context of municipal Health promotion (HP) practice. Accordingly, no attempt has been made to present methods, results or similar (as might be suggested by your evaluation that these parts "must be improved"), and rather, seek to make our focus and aim even clearer through the introduction.

Next, you "have difficulties in seeing how the authors articulate their article in regard to the literature. In other words, there are mentions of settings-based approach, of health in all policies, of health promotion, but not clear position about why salutogenesis is the most appropriate framework to contribute to structure collaboration process." We have tried to position us more clearly in respect to these overarching strategies within HP, and present a sound argument for why we think that applying and expanding Salutogenesis might make valuable contributions to the field of HP-practice (line 120-144). 

"Moreover, on the other side, authors refer sometimes to policies, sometimes to collaboration process, without explicitly mentioning what is the focus in the article and do not position their paper in the policy cycle or in regard to policy sciences, as how salutogenesis could be crossed and helpful in regard to this."

To meet this challenge, we seek to unravel the relationships between "policies" and "collaboration processes" in the introduction. We also included relevant literature from the field of policies and policy-making, to position us more clearly in regard of this field of knowledge, and illustrate how the discussed application of the theory might contribute to resolve challenges not only in respect to HP, but also, in respect to effective policy-making and -implementation.

"I would like to have seen more example of others studies questioning intersectorality, interdisciplinarity when its comes to local partnerships about health, and have more insights on how they are meaningful for the salutogenesis."

We have included more literature and prior research on policy-making and -implementation as well as local HP to meet this comment.

"Finally, the article lacks of structures and steps on how to use salutogenesis not just as asking questions or reflect on a process, but on how these questions should be used, at which stages, by whom, to provide full insights on its usefullness."

To meet this comment, a section about the structure of the article, unravelling which questions will be addressed where in the article, were included. Furthermore, we very much appreciate, and share, the eagerness to see whether these thoughts indeed yield benefits for practice- however, due to the conceptual nature of this article, we do hesitate to provide answers at this stage. Rather, we see this article as an attempt to make our thoughts clear and raise relevant questions, which we then will seek to address in later, empirical research.  

Reviewer 3 Report

This paper discusses an important approach oh health promotion that can have implications at munipality level. It is well supported and the authors provide illustrations (with the tables/boxes) that help the reader unrestand their points. 

Author Response

Thank you very much for your kind and positive review. We hope you enjoyed reading, and will recommend this paper to your colleagues!

Round 2

Reviewer 2 Report

The manuscript has really improved in terms of structure and soundness to the reader. Thank you to the authors.

Nevertheless, I still have concerns about how policy is treated in the present articles.

The title state:

Can the theory of Salutogenesis offer a framework to structure multi-sectoral collaboration for health promotion in municipalities?, but from my perspective, the aim of the article is to adress policy incoherence through theory of salutogenesis, which is not the same than multi-sectoral collaboration for health promotion.

Few examples:

LIne 64 to 67, All in all, local collaboration processes linked to the development, planning and implementation of HP policy are crucial part of developping HP policy and healthy society. This sentence is not precise in terms of what issue related to policy should this be adressed. Moreover, the link between HP policies and healthy society, as well as the role of municipality in this is not adressed either.

A second example, L80-87: Is it policy incoherence or policy implementation incoherence?
From my point of view, this is not the same. For example, you can have in two policy documents, two different types of health recommandations, coming both from different scientific societies, leading to confusion on which to respect, which is policy incoherence from my perspective. What you describe with (black lives matters" is the failure of policy implementation (as police do not respect the policy they have to implement).
From my point of view, this point is an example of the difficulty in the article to clarify where the reflection is situated. Is it policy framing? formulation? implementation?

From my point of view, if the title reflect what's in the article, then it should be presented as:  Can the theory of Salutogenesis offer a framework to enhance policy coherence in municipalities? 

Moreover, the authors did not adressed the question on what the theory of salutogenesis add as problem solving theory in comparison to health in all policies or healthy cities. Thoses approach claim for intersectorality, multi level implementation, positive vision of health too. So what is the added value or similarities and differences between these approaches, that the reader could benefit on ?

Finally, an interesting point raised on line 124 is the individual sense of coherence, that should be linked to societal and structures: how do the authors link it to policy development, implementation and evaluation?

 Finally, there is a lot of typos in the manuscript (e.g. Figure instead of Figur), that's need a proofreading of the final version.

Author Response

Dear reviewer, thank you, again, for your constructive feedback. We see your main point about a somewhat unclear focus, and have tried to clarify this focus throughout the text- as well as in the title, according to your recommendation (even if we made slight adaptions to clarify our focus even more).We hope that we with the below-presented changes have met your concerns, and made our focus and understandings clear. For more specific answers, see below.

Sincerely 

Specific answers:

"I still have concerns about how policy is treated in the present articles.

(...)from my perspective, the aim of the article is to adress policy incoherence through theory of salutogenesis, which is not the same than multi-sectoral collaboration for health promotion."

We see your point, and have clarified our focus throughout the paper as well as in the title. The main focus is on policy coherence- however, we highlight how policy incoherence can occur througout the whole policy-cycle, and maybe especially during local implementation. We therefor want to look at policy coherence also during local adotpion and implementation-which craves for extensive collab oration. We therefore also address challenges that can arise during collaboration- especially, during development, planning and implementation of policies.

This clarified focus implied that some paragraphgs have been changed substantially:

"LIne 64 to 67, All in all, local collaboration processes linked to the development, planning and implementation of HP policy are crucial part of developping HP policy and healthy society. This sentence is not precise in terms of what issue related to policy should this be adressed. Moreover, the link between HP policies and healthy society, as well as the role of municipality in this is not adressed either."

We have tried to clarify the municipalities role in HP (comapre line 28/29 and 36;37): We also clarify how HiAP is linked to HP policy, and how HiAP can be applied and realised in the local contect (line 30-40)

"A second example, L80-87: Is it policy incoherence or policy implementation incoherence?
From my point of view, this is not the same. For example, you can have in two policy documents, two different types of health recommandations, coming both from different scientific societies, leading to confusion on which to respect, which is policy incoherence from my perspective. What you describe with (black lives matters" is the failure of policy implementation (as police do not respect the policy they have to implement).
From my point of view, this point is an example of the difficulty in the article to clarify where the reflection is situated. Is it policy framing? formulation? implementation?"

This is an interesting point, which we discussed internally. However, as you might see from our above arguemnts, we think that challenges that arise during policy implementation can have major impact on polciy coherence- at least, from a stakeholder or popualtion-perspective (who might not know the policy uhnitl it is implemented- and which experiecne policy on how it is implemented). To amke our arguments more sound and meet you concerns, we have made this perspective explicit, and discuss how BLM can be linked to policy-coherence where the example is introduced (line 91-100).

"From my point of view, if the title reflect what's in the article, then it should be presented as:  Can the theory of Salutogenesis offer a framework to enhance policy coherence in municipalities? "

We have changed the title in line with your recommendation to better express our focus.

"Moreover, the authors did not adressed the question on what the theory of salutogenesis add as problem solving theory in comparison to health in all policies or healthy cities. Thoses approach claim for intersectorality, multi level implementation, positive vision of health too. So what is the added value or similarities and differences between these approaches, that the reader could benefit on ?"

We agree that these appraoches crave for "ntersectorality, multi level implementation, positive vision of health "- however, they do not provide pinpøoints towards how this can be achieved. We propose that Salutogenesis does provide some starting points to explore how it can be done- that is what we ant to explore in the discussion (compare line 159-169)

"Finally, an interesting point raised on line 124 is the individual sense of coherence, that should be linked to societal and structures: how do the authors link it to policy development, implementation and evaluation?"

We have tried to clarify our perspective in lines 137-140 and 151-156

" Finally, there is a lot of typos in the manuscript (e.g. Figure instead of Figur), that's need a proofreading of the final version."

We have proof-read, and aligned the language in the manuscript to english grammar and style consistently.

Round 3

Reviewer 2 Report

The authors have really make a great work of clarification and improvement in terms of referencing to policy science and other theoretical model. Thank you for these very interesting reflections and the opportunity to review this article.